# Using Landslide Statistical Index Technique for Landslide Susceptibility Mapping: Case Study: Ban Khoang Commune, Lao Cai Province, Vietnam

Long Nguyen Thanh [1], Yao-Min Fang [2,*], Tien-Yin Chou [2], Thanh-Van Hoang [2], Quoc Dinh Nguyen [1], Chen-Yang Lee [3], Chin-Lun Wang [3], Hsiao-Yuan Yin [3] and Yi-Chia Lin [3]

1 Vietnam Institute of Geosciences and Mineral Resources, 67 Chien Thang Rd, Van Quan Street, Ha Dong, Hanoi 100000, Vietnam

2 Geographic Information System Research Center, Feng Chia University, 100 Wenhwa Rd, Seatwen, Taichung City 40724, Taiwan

3 Soil and Water Conservation Bureau, Council of Agriculture, No 6, Guanghua Rd, Nantou City 540206, Taiwan

* Correspondence: frankfang@gis.tw; Tel.: +886-4-24516609

**Abstract:** Ban Khoang is a mountainous commune in Sa Pa district located in the central part of Lao Cai province, Vietnam. Landslides occur frequently in this area and seriously affect the local living conditions. To help the local authority in developing a landslide disaster action plan, the statistical index method for landslide susceptibility mapping is applied. As the result, the landslide susceptibility zonation (LSZ) map was created. The LSZ map indicates that areas of low, moderate, high and very high landslide susceptibility zones are, respectively, 20.3 km$^2$, 12.4 km$^2$, 15.4 km$^2$, and 5.2 km$^2$; most of the observed landslide areas that are well predicted belong to high or very high landslide susceptibility classes. In detail, 80% observed landslide areas and 78.57% number of observed landslides were well predicted, and the area (AUC) under the receiver operating characteristic (ROC) curve obtained 80.3%. Hence, the high and very high landslide susceptibility classes in the LSZ map can be considered highly believable, and the LSZ map will be reliable to use in the practice.

**Keywords:** natural hazards; landslide; susceptibility; GIS; Vietnam

## 1. Introduction

Ban Khoang is a mountainous commune in Sa Pa district, Lao Cai province of Vietnam, where these landslides occur regularly (Figure 1). In particular, a vast landslide happened in Can Ho A village, Ban Khoang commune in September 2013, causing 14 people loss and severe property damage. Hence, predicting landslide hazards is very important for the inhabitants and local administration of Ban Khoang commune to mitigate landslide damage in this area.

According to the result of a nationwide project "Investigation, assessment and geohazards susceptibility zonation in mountainous areas of Vietnam" [1] recently, Ban Khoang is one of 200 communes with highest level of landslides susceptibility in Vietnam.

Therefore, the LSZ mapping will be very necessary and helpful for local authorities and people in landslide hazard prevention and mitigation, as well as developing a landslide action plan. In addition, the LSZ map will be a technical foundation for practical activities relevant to setting up landslide early warning systems.

The most straightforward initial approach to any study of landslide hazards is the compilation of a landslide inventory and analyzing the relationship with different causative factors to predict landslide-prone areas [2]. In Ref. [3], Carrara (1983) introduced the so-called statistical approach for landslide hazard assessment. This technique has been widely employed and has become one of the most popular approaches for landslide hazard

assessment worldwide. Combinations of factors that have led to landslides in the past are identified statistically, and quantitative predictions are made for areas currently free of landslides but with similar conditions. Since then, many other statistical approaches have been proposed and used in landslide susceptibility mapping and analyses.

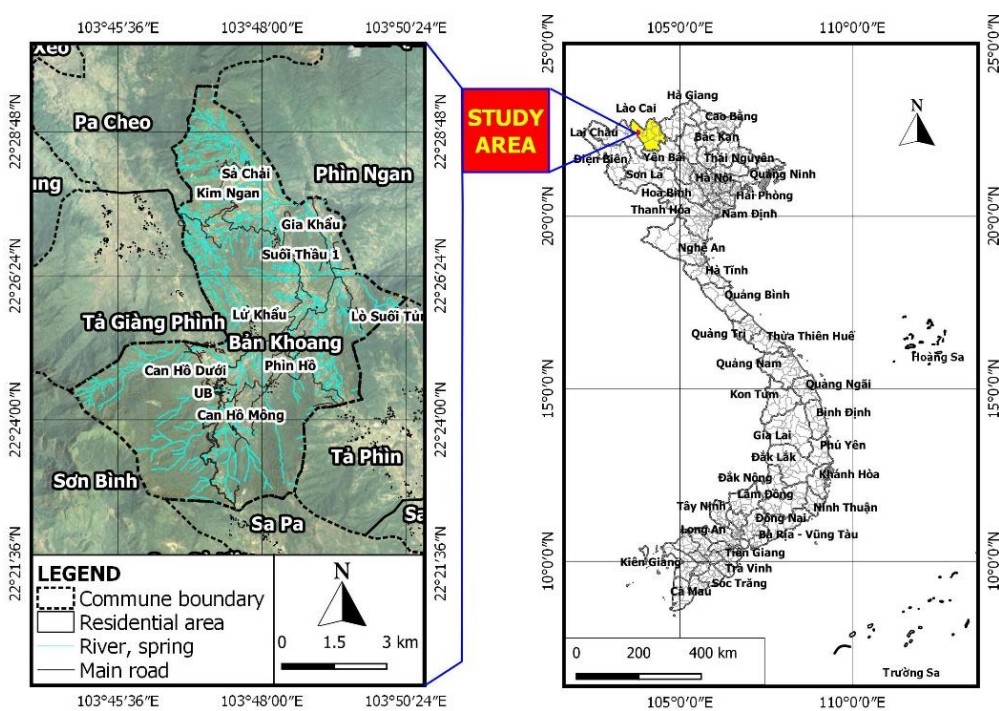

**Figure 1.** In this study area, 28 landslides, covering 0.262 km$^2$ (Figure 2), were identified (from 2012 up to May 2022) based on (1) field recognizance to investigate landslide occurrences, (2) collection of historical literature on landslides, and (3) interpretation of available multi-serial google images coupled with field verification.

Basically, statistical landslide susceptibility approaches are based on related spatial information on past landslide activities (i.e., landslide presence/absence) to static geoenvironmental factors (e.g., topography, geology, geomorphology, land use, fault density, soil, and drainage density) using statistical techniques. In Ref. [4], Steger et al. (2016) commented that the generated empirical relation, commonly expressed as a relative susceptibility score, is then applied to each spatial unit of an area (e.g., grid cell, and slope unit) [5–7]. The validation of spatial predictions is commonly evaluated by interpreting inventory-based predictive performance estimates [8–10].

It is obvious that the landslide inventory is a vital component to obtaining high-quality statistical landslide susceptibility models because most analysis steps are dependent on a correct representation of past landslide occurrences [4,9,11–14].

Several studies compared statistical landslide susceptibility models produced from heterogeneous inventories [4,15–19]. However, a differentiated evaluation of the propagation of potential inventory-based errors into landslide susceptibility models was hampered due to the practical inseparability of positional accuracy and inventory completeness as well as the lack of truly accurate reference inventories.

There are many previous works using the statistical approaches for landslide susceptibility assessment (e.g., methods of statistical index, certainty factor, probability, weight of evidence modeling, and logisitic regression). However, the selection of input parameters or causative factors for landslide susceptibility mapping, the method for landslide susceptibility mapping and landslide susceptibility index classification are still confused between many studies.

The statistical index method is considered the simplest and quantitatively suitable method for statistical approaches for landslide susceptibility mapping. However, it has been adopted by various researchers [19–27].

Therefore, in this study, the statistical index method is applied for landslide susceptibility analyses of Ban Khoang commune in Sa Pa district, Lao Cai province of Vietnam. The research result will play an important role for landslide hazard prevention and mitigation in this mountainous commune in Vietnam.

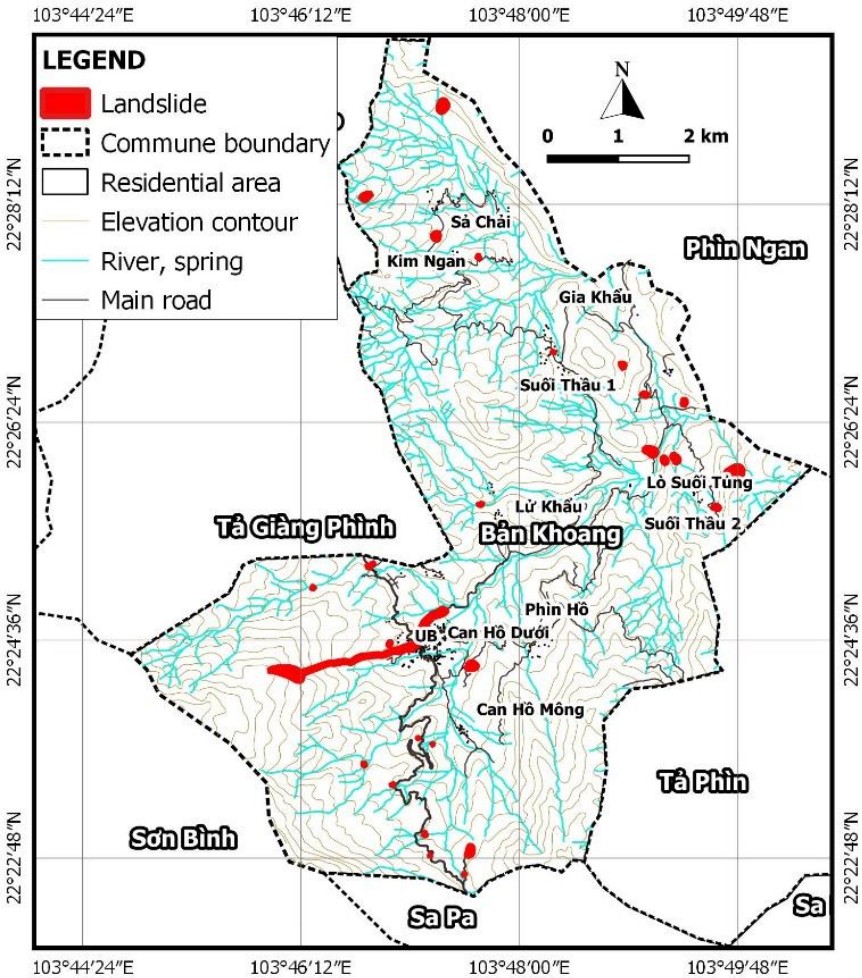

**Figure 2.** Map of landslide inventory in the study area.

## 2. Landslide Inventory

The study area, Ban Khoang commune selected for assessment of landslides susceptibility (Figure 1) is about 53.3 km$^2$.

The average size of the landslides in the study area is approximately 9369 m$^2$, but the details about width, depth, types, or causes of some landslides were not identified. Some pictures of landslide inventory are displayed in Figure 3.

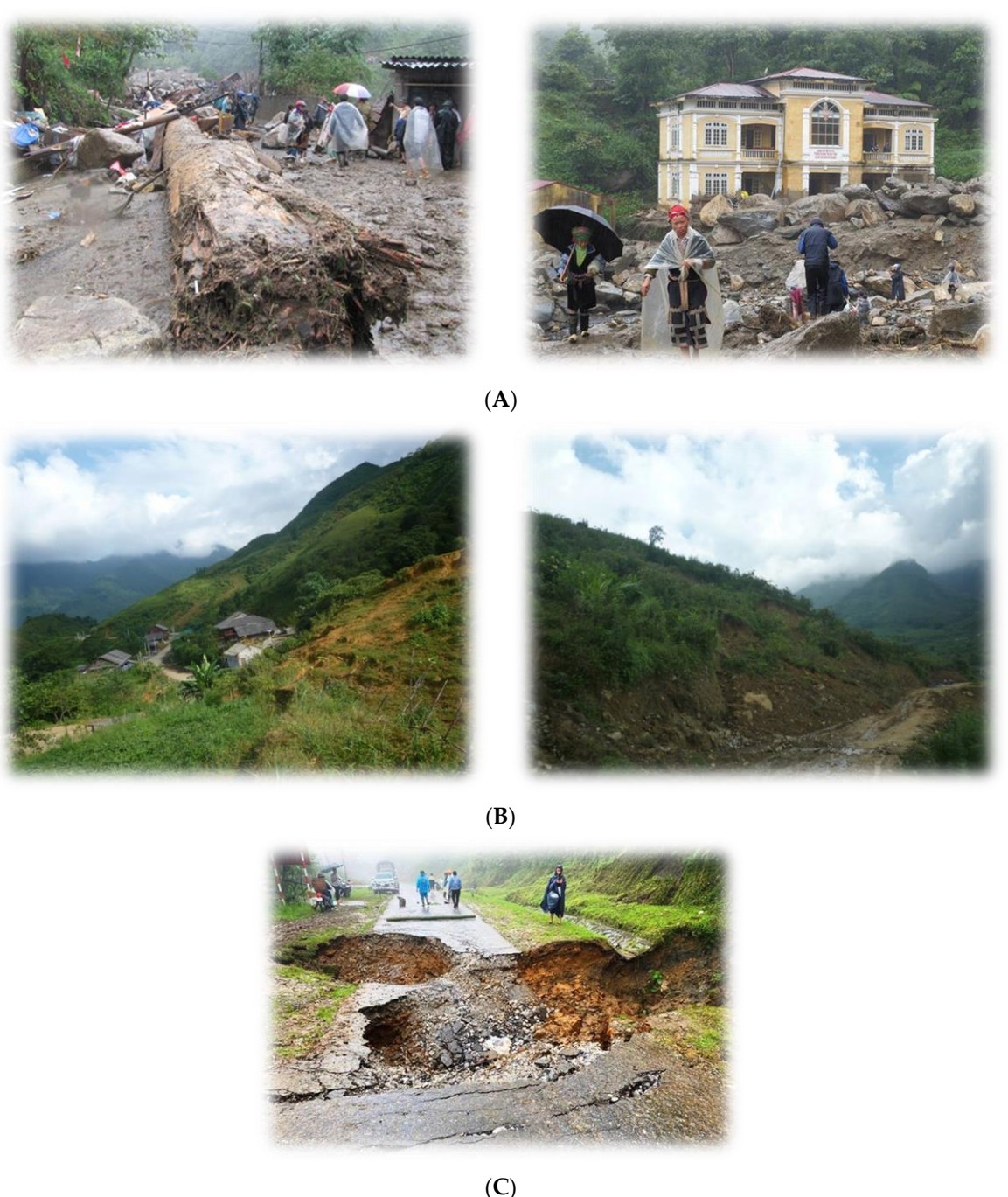

**Figure 3.** Some landslide pictures in Ban Khoang commune, Sa Pa district, Lao Cai province of Vietnam. (**A**) Landslide as debris flow occurred near hospital of Ban Khoang commune in 2013. (**B**) Landslides near provincial road 155 in Ban Khoang commune. (**C**) Landslide on the provincial road 155, the section closed to Can Hồ B village in Ban Khoang commune on 22 May 2022.

## 3. Landslide Causative Factors

The selection of causative factor maps for landslide susceptibility should be considered carefully based on relevance, availability and scale attributes. These are cumbersome in Vietnam, as systematic studies and inventories of spatial characteristics and land cover features have only been initiated recently by different government institutions. Therefore, such data are often lacking, incomplete, or on a scale that is not useful for scientific purposes, especially in remote and rural regions as the present study area. Ham-

pered by such constraints, eight digital causative factors map for landslide analysis could be developed:

- Topography is intrinsically associated with landslides by slope gradient and other factors, such as weathering, precipitation, soil thickness, etc. Hence, topography strongly affects landslides [28,29]. Ban Khoang is a mountainous area where the microclimate is quite predominant. Hence, the aspect is considered an indirect landslide causative factor in this study. A digital elevation map (DEM) of the study area with a pixel size of 10 m by 10 m was obtained by using inverse distance weighted interpolation in QGIS 3.6 from elevation points and contours of a topographic map, scale 1:10,000, published by the Cartographic Publishing House, Vietnamese Ministry of Natural Resources and Environment (2019). Then the aspect map of Ban Khoang commune (Figure 4A) was developed based on the Aspect tool inside QGIS 3.6 software.

- In most landslide studies, slope gradient is considered a principal causative or triggering factor. A slope map was derived from the DEM using the slope function tool of QGIS 3.6. The slope map is in the form of a raster map with the same 10 m pixel size as the DEM, but was converted to vector by separating the slope angles into six classes: (1) flat-gentle slope (<5°), (2) fair slope (5–15°), (3) moderate slope (15–25°), (4) fairly moderate slope (25–35°), (5) steep slope (35–45°), and (6) very steep slope (>45°). The map of slope classes of Ban Khoang commune is displayed in Figure 4B.

- Geology and slope instability are strongly associated [30,31]. Hence, a geological map of Ban Khoang (Figure 4C) was derived from the map of geology and mineral resources of the Lao Cai sheet group, scale 1:50,000 by Lap et al. (2003) [32]. Figure 4C displays the distribution of geological classes in Ban Khoang commune in Sa Pa district, Lao Cai province of Vietnam.

- Geomorphology is considered an essential factor related to landslide occurrence in the study area. Based on the analyses of the topological characteristics, geological structures, neotectonic movements, and morphometries, six geomorphological units can be identified in the study area by [33] (Figure 4D).

- Soil is an essential factor of slope instability in many settings [34,35]. A digital map of soil was derived from previous work in Lao Cai province carried out by the National Institute of Agriculture Planning and production (2019), identifying three types of soil mechanics in the study area, i.e., (1) outcrop, (2) reddish-yellow humus soil on claystone, and (3) reddish-yellow humus soil on magma rocks (Figure 4E). The soil depth map (Figure 4F) was derived based on the soil depth information based on the map of soil mechanics.

- Neotectonics contribute to slope instability by fracturing, faulting, jointing, and deforming foliation structures [36,37]. For this study, faults were extracted from the map of geology and mineral resources scale 1:50,000. Additionally, lineaments were interpreted from free available Landsat 8 captured by NASA in 2020. The fault and lineament density was calculated as the total length of faults and lineament per 1 km$^2$ (See Figure 4G).

- Studies have shown that the proximity to drainage axes with intensive gully erosion is an important factor controlling the occurrence of landslides [38,39]. A map of river density was derived on the basis of the digitizing river and stream courses on the topographic map and interpolation in QGIS software (version 3.6). A map of the river density class (Figure 4H) was created by subdividing the river density range values into five classes: (1) <1000 m/km$^2$, (2) 1000–2000 m/km$^2$, and (3) 2000–3000 m/km$^2$, (4) 3000–4000 m/km$^2$, and (5) >4000 m/km$^2$.

- Vegetation augments slope stability primarily in two ways: (1) by removing soil moisture through evapotranspiration and (2) by providing root cohesion to the soil mantle [40]. A land-use map was obtained from the land-use map of Lao Cai published by the land administration department of the Ministry of Natural Resources and Environment, 2019 [41]. The land use composed of 10 land-use classes is displayed in Figure 4I.

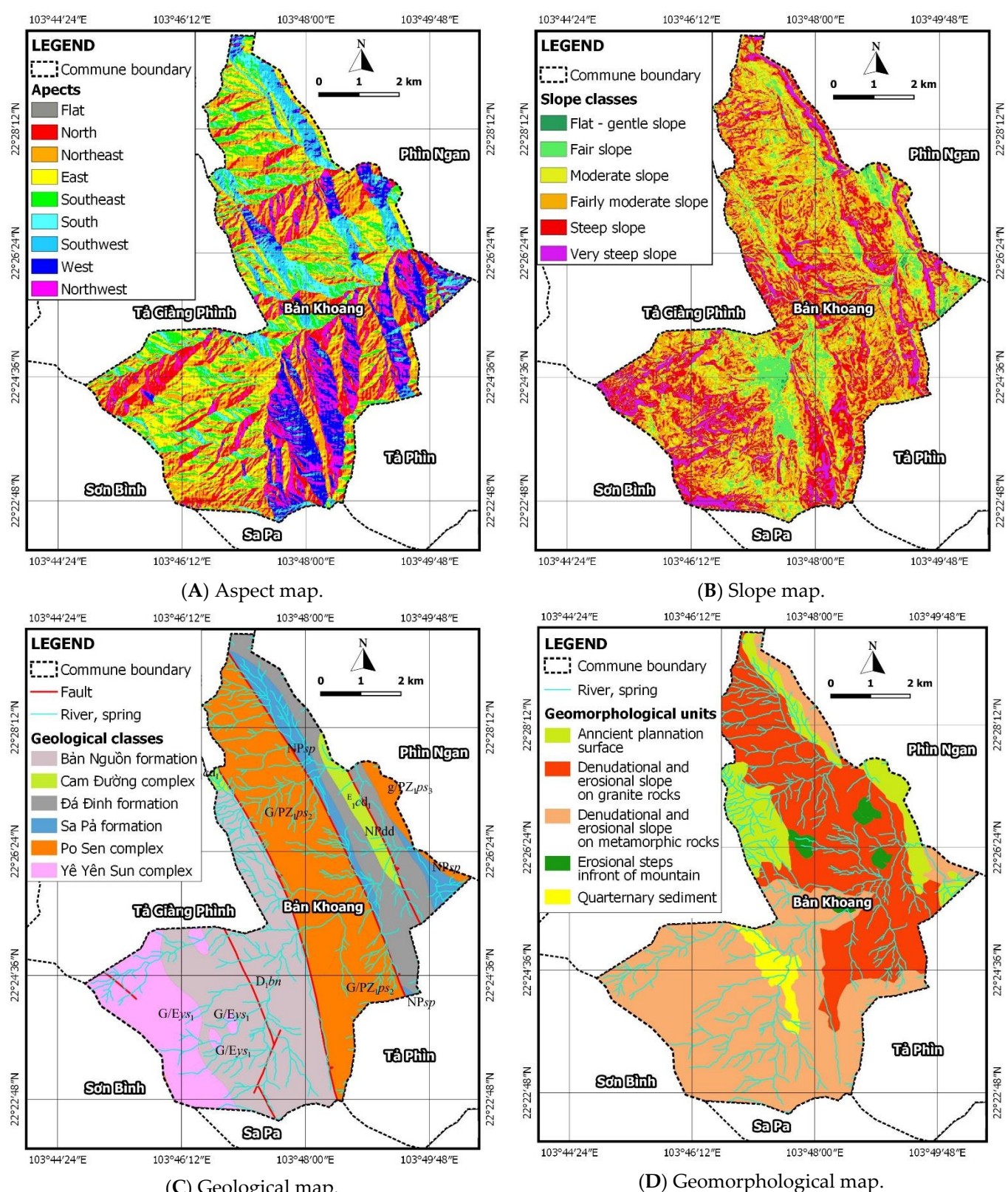

(**A**) Aspect map.

(**B**) Slope map.

(**C**) Geological map.

(**D**) Geomorphological map.

**Figure 4.** *Cont.*

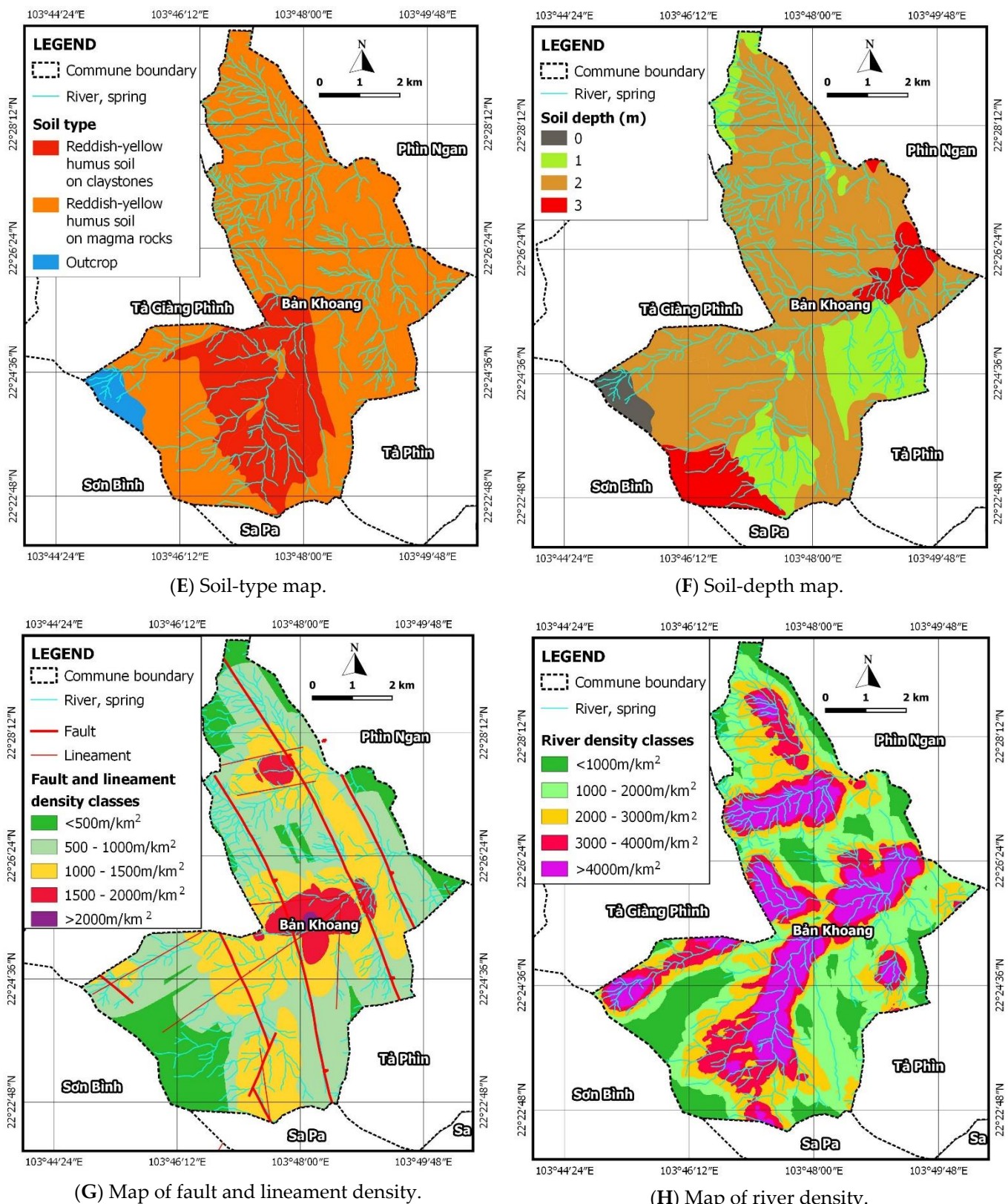

(**E**) Soil-type map.

(**F**) Soil-depth map.

(**G**) Map of fault and lineament density.

(**H**) Map of river density.

**Figure 4.** *Cont*.

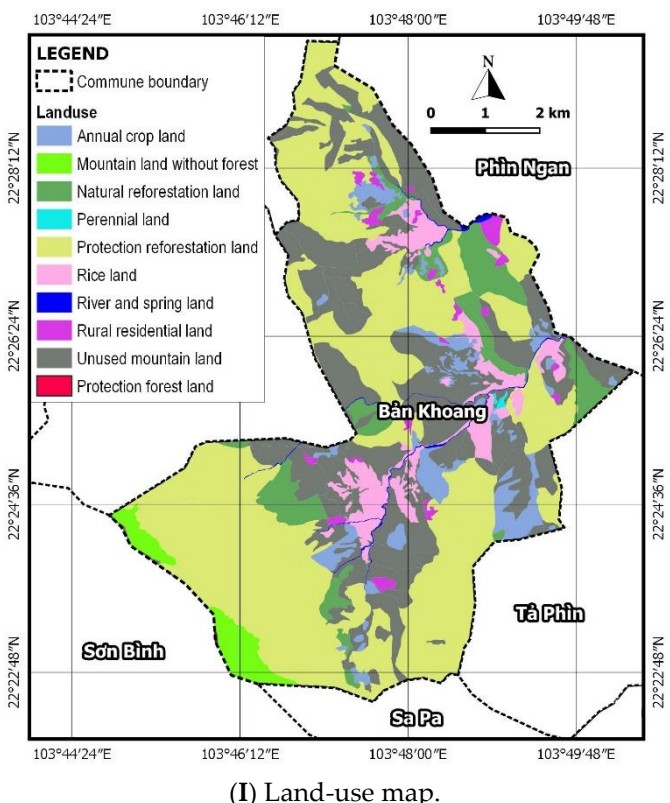

(**I**) Land-use map.

**Figure 4.** Landslide causative parameters for landslide susceptibility mapping of Ban Khoang commune.

## 4. Method for Landslide Susceptibility Analysis

The statistical index method is a bivariate statistical technique introduced by van Westen in 1997 [20] for landslide susceptibility analyses. Other researchers, such as Gebremedhin et al., 2021 [21], Mandal et al., 2018 [22], Wu et al., 2017 [23], Wang et al., 2016 [24], Dieu et al., 2011 [25], Long, 2008 [19], Cevik and Topal, 2003 [27], and Oztekin and Topal, 2005 [26], also applied this technique and termed it the statistical index method. In the statistical index method, a weight value for a parameter class is defined as the natural logarithm of the landslide density in the class divided by the landslide density in the entire map [20].

$$W_{ij} = \ln\left(\frac{f_{ij}}{f}\right) \tag{1}$$

where $W_{ij}$ is the weight of a class i of parameter j, $f_{ij}$ the landslide density within the class i of parameter j, and f the landslide density within the entire map. Hence, the statistical index method is based on statistical correlation of the landslide inventory map with attributes of different parameter maps. The $W_{ij}$ value in Equation (1) is only calculated for classes that have landslide occurrences. If there are no landslide occurrences in a parameter class, the $W_{ij}$ will be assigned to zero [20,30]. This also means that the parameter class having no landslide occurrences will have no correlation with the landslide inventory. Hence, it does not influence the calculation of the landslide susceptibility index.

In this study, nine landslide causative factors, i.e., (1) slope, (2) geology, (3) geomorphology, (4) soil depth, (5) soil type, (6) land use, (7) fault and lineament density, and (8) river density (Figure 4), were used as the layer input for landslide susceptibility index mapping. The workflow for landslide susceptibility mapping in Ban Khoang commune is shown in Figure 5. Every parameter map is crossed with the landslide map, and the density of the landslide in each class is calculated. The distribution of landslides for various data layers and weight wij values are shown in Table 1. The distribution of landslides for

various data layers, weight of class ($W_{ij}$) of all the causative factors in the study area is displayed in Table 2.

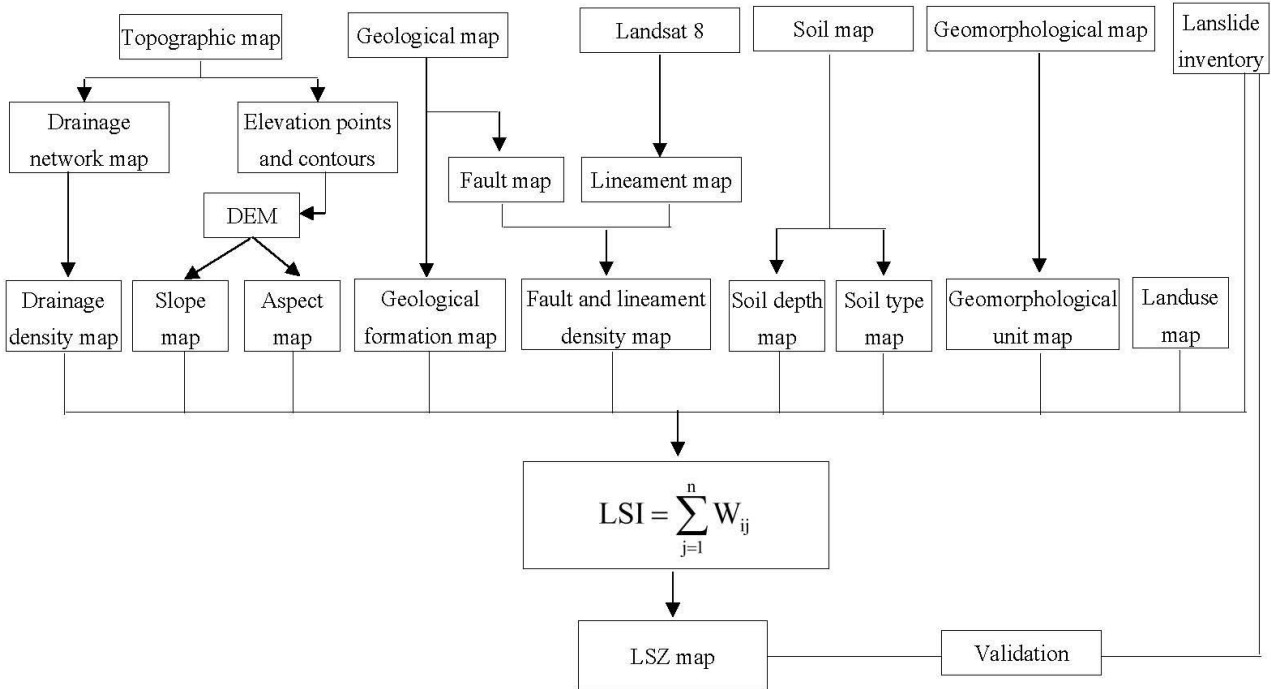

**Figure 5.** Work flow for landslide susceptibility mapping in the study area.

**Table 1.** Distribution of landslides for various data layers, weight of class ($W_{ij}$) of all causative factors in the study area.

| Landslide Causative Factors | Landslide Occ. Pixels | % Occ. | No. of Pixels in Domain | % Domain | $W_{ij}$ |
|---|---|---|---|---|---|
| **Slope** | | | | | |
| <5° | 11 | 0.42 | 2798 | 0.52 | −0.2261 |
| 5–15° | 537 | 20.43 | 45,087 | 8.46 | 0.8823 |
| 15–25° | 580 | 22.07 | 114,691 | 21.51 | 0.0257 |
| 25–35° | 508 | 19.33 | 182,665 | 34.26 | −0.5723 |
| 35–45° | 609 | 23.17 | 138,605 | 25.99 | −0.1149 |
| >45° | 383 | 14.57 | 49,356 | 9.26 | 0.4539 |
| **Fault and lineament density** | | | | | |
| <500 m/km$^2$ | 348 | 13.24 | 81,710 | 15.32 | −0.1461 |
| 500–1000 m/km$^2$ | 1103 | 41.97 | 235,319 | 44.13 | −0.0502 |
| 1000–1500 m/km$^2$ | 1170 | 44.52 | 178,785 | 33.53 | 0.2835 |
| 1500–2000 m/km$^2$ | 7 | 0.27 | 35,889 | 6.73 | −3.2296 |
| >2000 m/km$^2$ | 0 | 0.00 | 1499 | 0.28 | 0.0000 |
| **River density** | | | | | |
| <1000 m/km$^2$ | 758 | 28.84 | 86,095 | 16.15 | 0.5802 |
| 1000–2000 m/km$^2$ | 603 | 22.95 | 135,804 | 25.47 | −0.1044 |
| 2000–3000 m/km$^2$ | 684 | 26.03 | 113,416 | 21.27 | 0.2018 |
| 3000–4000 m/km$^2$ | 250 | 9.51 | 96,950 | 18.18 | −0.6478 |
| >4000 m/km$^2$ | 333 | 12.67 | 100,937 | 18.93 | −0.4014 |
| **Soil depth** | | | | | |
| 0 m | 0 | 0.00 | 11,686 | 2.19 | 0.0000 |
| 1 m | 175 | 6.66 | 94,253 | 17.68 | −0.9763 |
| 2 m | 2314 | 88.05 | 376,835 | 70.67 | 0.2198 |
| 3 m | 139 | 5.29 | 50,428 | 9.46 | −0.5812 |

**Table 1.** *Cont.*

| Landslide Causative Factors | Landslide Occ. Pixels | % Occ. | No. of Pixels in Domain | % Domain | $W_{ij}$ |
|---|---|---|---|---|---|
| **Soil type** | | | | | |
| Reddish-yellow humus soil on magma rocks | 1629 | 61.99 | 401,940 | 75.38 | −0.1957 |
| Outcrop | 0 | 0.00 | 11,686 | 2.19 | 0.0000 |
| Reddish-yellow humus soil on claystone | 999 | 38.01 | 119,576 | 22.43 | 0.5277 |
| **Geomorphology** | | | | | |
| Ancient planation surface | 93 | 3.54 | 59,101 | 11.08 | −1.1417 |
| Denudational and erosional slope on metamorphic rocks | 1597 | 60.77 | 193,797 | 36.35 | 0.5140 |
| Denudational and erosional slope on granite rocks | 489 | 18.61 | 256,604 | 48.13 | −0.9502 |
| Quaternary sediment | 427 | 16.25 | 9101 | 1.71 | 2.2533 |
| Erosional steps in front of mountain | 22 | 0.84 | 14,599 | 2.74 | −1.1850 |
| **Geology** | | | | | |
| Sa Pả formation | 262 | 9.97 | 32,774 | 6.15 | 0.4836 |
| Cam Đường formation | 169 | 6.43 | 15,573 | 2.92 | 0.7893 |
| Yê Yên Sun complex | 571 | 21.73 | 69,501 | 13.03 | 0.5110 |
| Po Sen complex | 163 | 6.20 | 183,843 | 34.48 | −1.7154 |
| Đá Đinh formation | 312 | 11.87 | 64,195 | 12.04 | −0.0140 |
| Bản Nguồn formation | 1151 | 43.80 | 167,316 | 31.38 | 0.3334 |
| **Landuse** | | | | | |
| Protection reforestation land | 1089 | 41.44 | 252,972 | 47.44 | −0.1353 |
| Rice land | 241 | 9.17 | 29,879 | 5.60 | 0.4926 |
| Annual crop land | 2 | 0.08 | 33,860 | 6.35 | −4.4242 |
| Natural reforestation land | 585 | 22.26 | 39,883 | 7.48 | 1.0906 |
| Rural residential land | 16 | 0.61 | 8229 | 1.54 | −0.9302 |
| Unused mountain land | 682 | 25.95 | 152,292 | 28.56 | −0.0958 |
| Mountain land without forest | 0 | 0.00 | 12,683 | 2.38 | 0.0000 |
| Perennial land | 0 | 0.00 | 619 | 0.12 | 0.0000 |
| River and spring land | 13 | 0.49 | 2660 | 0.50 | −0.0085 |
| Protection forest land | 0 | 0.00 | 125 | 0.02 | 0.0000 |
| **Aspect** | | | | | |
| Flat | 0 | 0.00 | 24 | 0.00 | 0.0000 |
| North | 422 | 16.06 | 80,933 | 15.18 | 0.0563 |
| Northeast | 904 | 34.40 | 103,501 | 19.41 | 0.5722 |
| East | 586 | 22.30 | 102,496 | 19.22 | 0.1484 |
| Southeast | 285 | 10.84 | 64,916 | 12.17 | −0.1157 |
| South | 100 | 3.81 | 40,339 | 7.57 | −0.6872 |
| Southwest | 102 | 3.88 | 33,324 | 6.25 | −0.4764 |
| West | 58 | 2.21 | 54,362 | 10.20 | −1.5303 |
| Northwest | 171 | 6.51 | 53,307 | 10.00 | −0.4295 |

**Table 2.** Distribution of landslides for various data layers, weight of class ($W_{ij}$) of all causative factors in the study area.

| Landslide Causative Factors | $LSI_{Min}$ | $LSI_{Max}$ | $LSI_{Range}$ | $LSI_{Dev}$ |
|---|---|---|---|---|
| Slope | −0.5723 | 0.8823 | 1.4546 | −0.5183 |
| Fault and lineament density | −3.2296 | 0.2835 | 3.5131 | −1.4628 |
| River density | −0.6478 | 0.5802 | 1.2280 | −0.4851 |
| Soil depth | −0.9763 | 0.2198 | 1.1961 | −0.5453 |
| Soil type | −0.1957 | 0.5277 | 0.7234 | −0.3742 |
| Geomorphology | −1.1850 | 2.2533 | 2.5047 | −0.9107 |
| Geology | −1.7154 | 0.7893 | 2.5047 | −0.9107 |
| Land use | −4.4242 | 1.0906 | 5.5147 | −1.5015 |
| Aspect | −1.5303 | 0.1484 | 2.1025 | −0.6039 |

From Tables 1 and 2, it can be noted the following:

- For the slope factor, there is an obvious distinction between classes with slope angles 5–15° and >45° compared to other classes. This indicates that landslides in the study area are mainly occurring in areas with slope angles 5–15° and >45°.
- The class of fault density of 1000–1500 m/km² has the highest $W_{ij}$ value (0.2835) compared to the remaining classes from all causative factors; hence, it has the highest impact on landslides in the study area.
- Cam Đường formation ($W_{ij}$ = 0.7893) are distinctly more favorable for landslides compared to the other geological formations ($W_{ij} \leq 0.5110$).
- For the geomorphological factor, denudational and erosional slope on metamorphic rocks, Quaternary sediment, also favor landslides.
- For the land-use factor, natural reforestation land is most favorable for landslide occurrence. Other classes seem to have very little or no influence for landslides.

All $W_{ij}$ layers for the different causative factors were constructed with QGIS 3.6 software. Next, these were summed up to obtain a resultant landslide susceptibility index map.

$$LSI = \sum_{j=1}^{n} W_{ij} \tag{2}$$

where LSI is the landslide susceptibility index and n the number of parameters.

As the result, the LSI map of Ban Khoang commune was developed and is displayed in Figure 6.

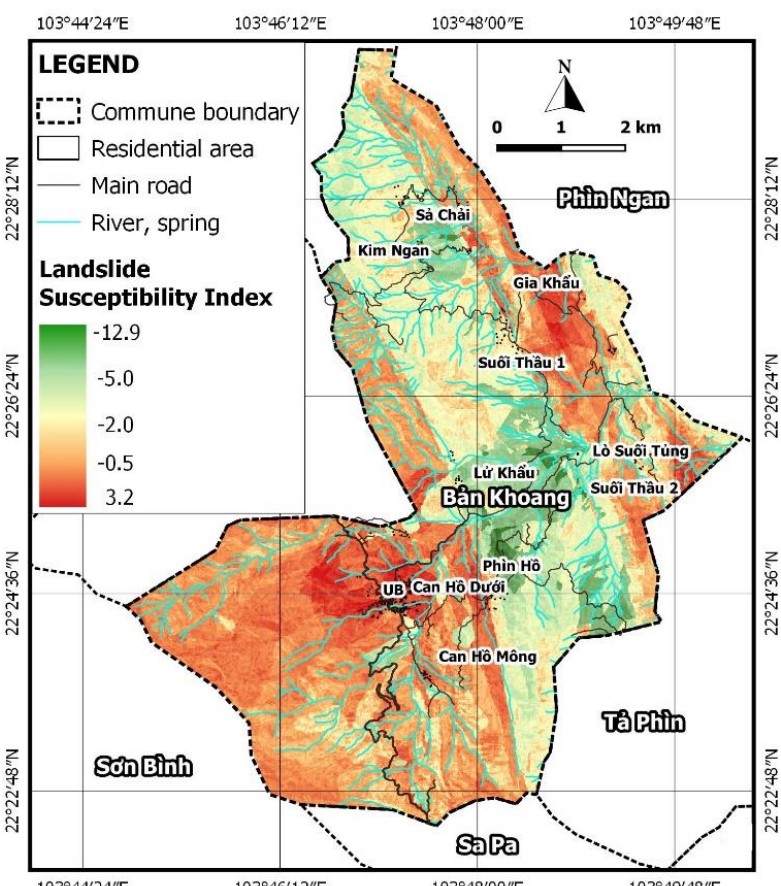

**Figure 6.** Map of landslide susceptibility index of Ban Khoang commune.

## 5. Results and Discussion

Classifier methods that have been used in landslide classification are manual classification [26,38,42–48], equal interval classification [49,50], standard deviation classification [51–55]. However, the authors usually do not explain the reasons for using a certain method in previous works.

In this study, the manual classifier method was used to reclassify the LSI values into four different susceptibility zones, according to the classification method that was proposed by Galang (2004) [56]. The susceptibility classes are low, moderate, high, and very high. Ideally, the classification method should satisfy the principle that higher landslide susceptibility classes should capture more or most landslide occurrences. Therefore, it is assumed that the expected number of observed landslide occurrences within a higher landslide susceptibility class equals two times the expected numbers in the next lower landslide susceptibility class. Hence, the expected numbers of observed landslide occurrences in the very high landslide susceptibility class equals two times the expected numbers in the high landslide susceptibility class, and so on. Based on this rule, it can be inferred that the expected percentages of observed landslide occurrences in the low, moderate, high, and very high landslide susceptibility classes are 6.7%, 13.3%, 26.7%, and 53.3% respectively.

Hence, the procedure is as follows. The landslide occurrence map is compared to the LSI map, and the cumulative percentage of observed landslide values versus ranked LSI values is calculated as shown in Figure 6. Three cut-off percentages of observed landslide occurrence in the cumulative curve are used to identify the four landslide susceptibility classes. It is 6.7% for separating the low from the moderate class, 20% for separating the moderate from high class, and 46.7% for separating the high from the very high class, as shown in Figure 7.

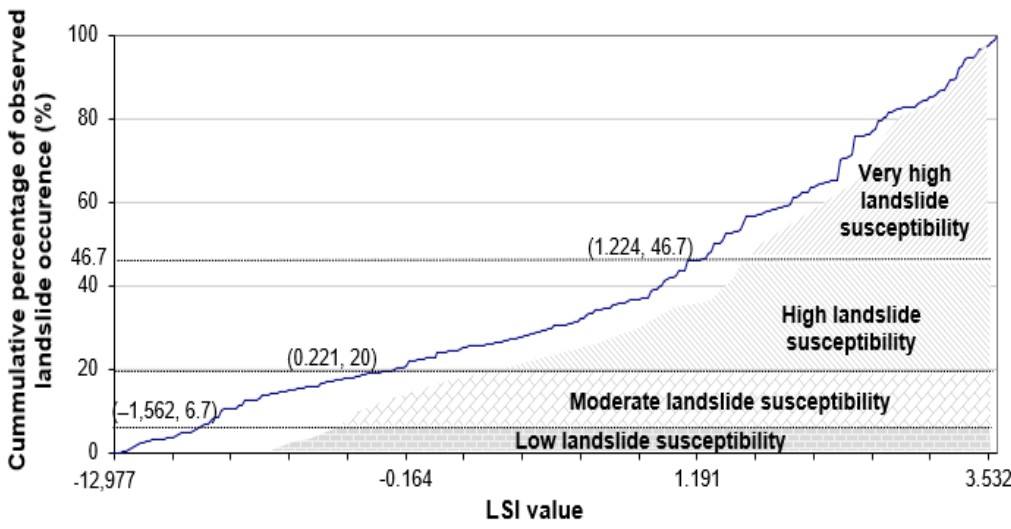

**Figure 7.** Cumulative percentage of observed landslide occurrence against LSI values.

As the result, the final map of landslide susceptibility zonation is shown in Figure 8. The statistical index shows that areas of low, moderate, high and very high landslide susceptibility zones are, respectively, 20.3 km$^2$ (38.0%), 12.4 km$^2$ (23.3%), 15.4 km$^2$ (28.9%), and 5.2 km$^2$ (9.8%).

In addition, to minimize the damage caused by natural disasters caused by climate change to people in Lao Cai province, Taiwan's Soil and Water Conservation Bureau (SWCB) and the Vietnam Institute of Science and Mineral Geology (VIGMR) have built a landslide monitoring station in Ban Khoang commune, Lao Cai in November 2019.

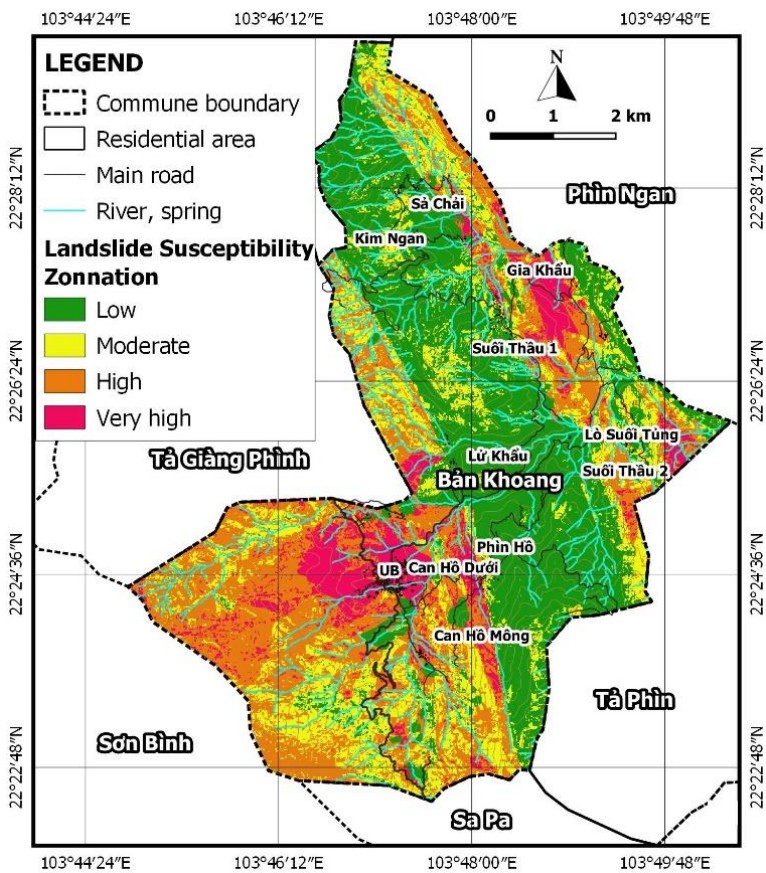

**Figure 8.** Map of landslide susceptibility zonation of Ban Khoang commune.

According to a survey by the VIGMR, Ban Khoang commune (Lao Cai province) is the area most at risk of landslides in Lao Cai province. At the same time, it is also the place with the highest risk of landslides in Vietnam. Therefore, installing a real-time landslide monitoring station in these two areas is essential.

Within the framework of international cooperation among the SWCB, GIS.FCU (the Geography information System Research Center of Taiwan Feng Chia University), VIGMR, WeatherPlus company (Former is AgriMedia), the projects "Study, develop a pilot debris flow early warning system in real time for mountainous areas of Vietnam" and "Study, develop a pilot debris flow early warning system in real time for mountainous areas of Vietnam" were applied to Ban Khoang area, Sa Pa town, Lao Cai province.

The real-time landslide early warning system deployed and installed in Ban Khoang includes a series of sensors, such as geophone, water level, tensiometer sensor, infrared cameras, and auto-rain gauges installed in three areas (Figure 9) (upstream (Figure 10) and midstream (Figure 11), and downstream (Figure 12)) to observe and record changes in weather conditions, such as precipitation, geology (by geogphone), hydrology (flow, water level) and the surface movement of liquid mud, soil and rock. All data are collected and processed on site (Figure 13) by the Data Processing Center.

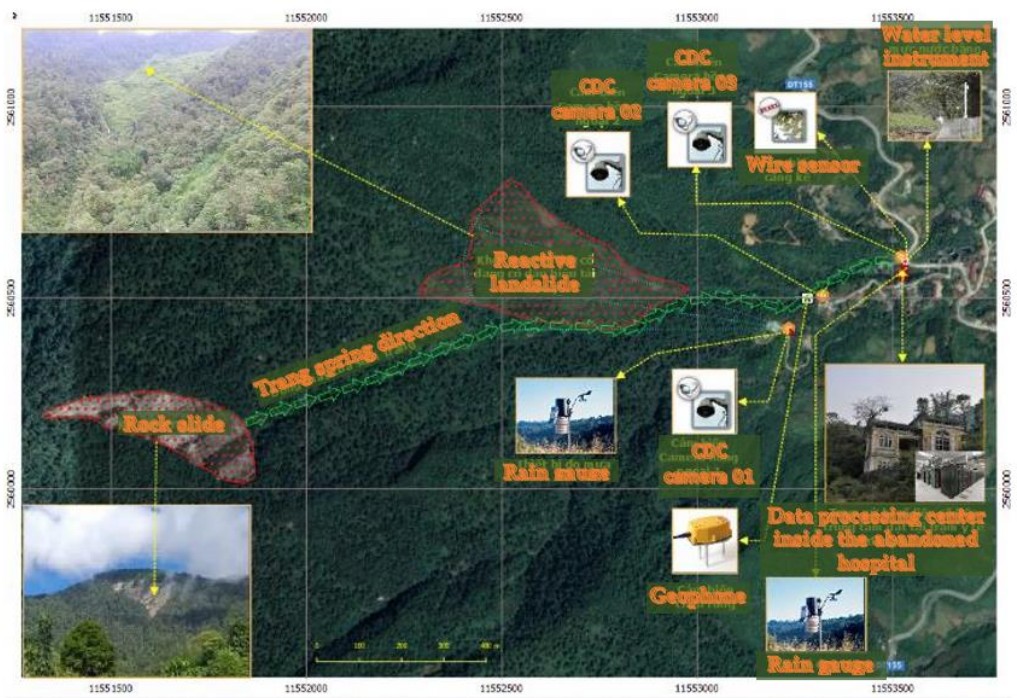

**Figure 9.** Real-time landslide monitoring station system in Ban Khoang, Lao Cai.

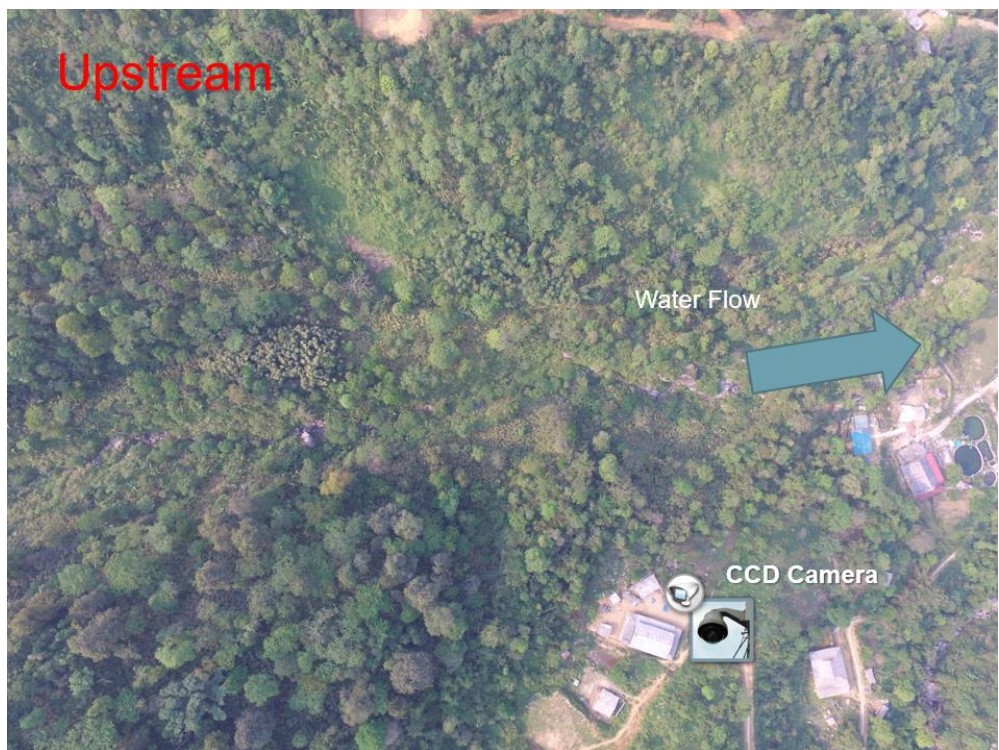

**Figure 10.** Upstream of the real-time landslide monitoring station.

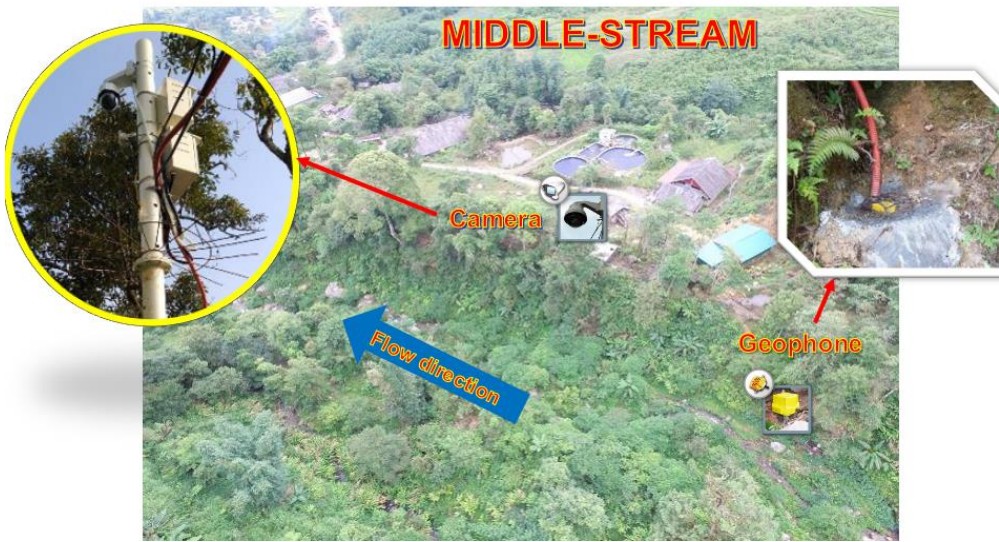

**Figure 11.** Middle stream of the real-time landslide monitoring station.

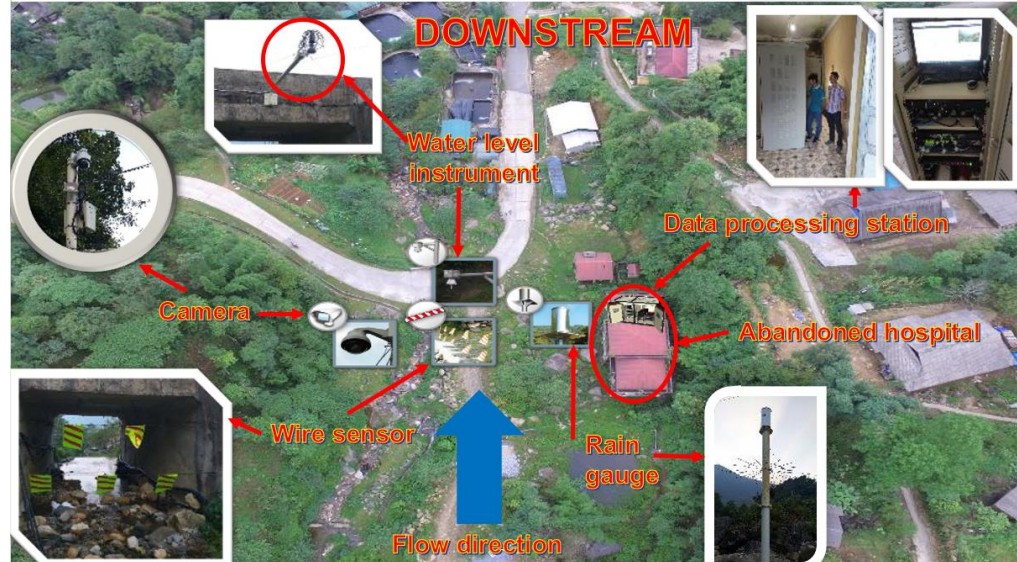

**Figure 12.** Downstream of the real-time landslide monitoring station.

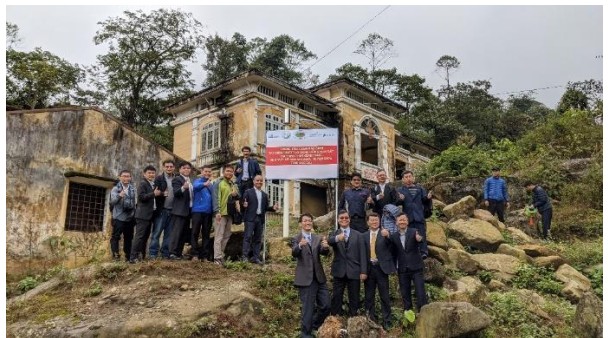
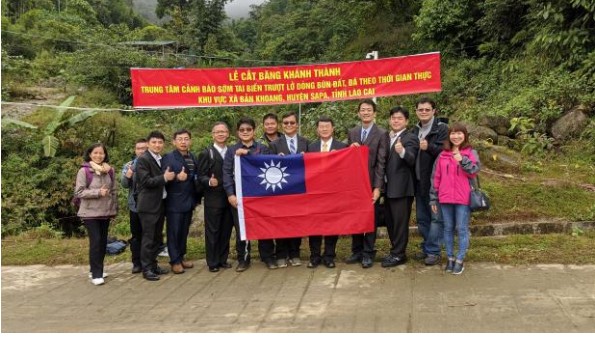

**Figure 13.** On-site station at the downstream of the real-time monitoring station.

## 6. Validation of Landslide Susceptibility Map

The final map of landslide susceptibility zonation for the study area is shown in Figure 8, and the area percentages of landslide susceptibility classes and posterior landslide susceptibility of these classes in the final LSZ map are shown in Figure 14.

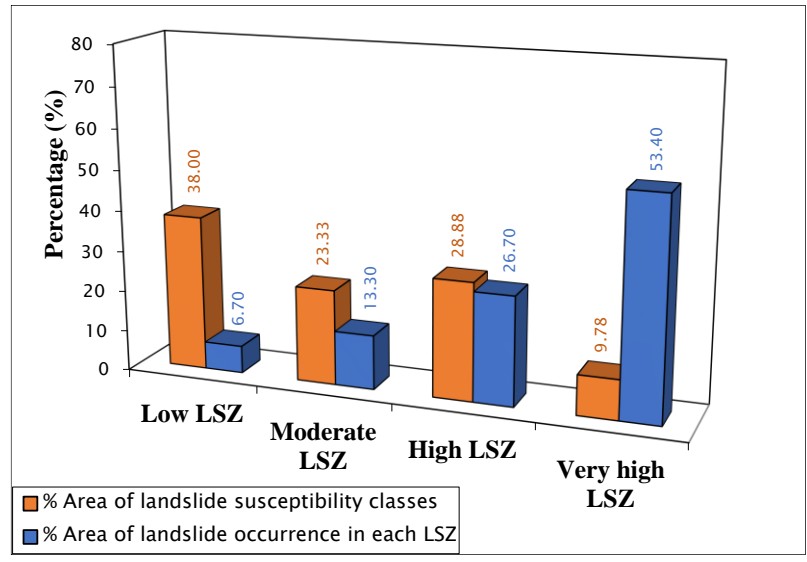

**Figure 14.** Area percentage of LSZ classes and the observed landslide accumulation in each LSZ class.

The accuracy of the final LSZ map is evaluated based on the observed landslides. First, Figure 13 shows that 80% observed landslide areas belonging to very high and high LSZ classes. Secondly, the final LSZ map is checked by overlaying it with the observed landslide map. In addition, As shown in Figure 14, there are various possibilities of different LSZs coinciding with a landslide polygon. Because in the inventory of the observed landslide, no distinction was made between the initiation part of the landslide and the areas of debris or flows, there can be no complete correspondence between the LSZ classes (Figure 15) that blue line is landslide area, and the complete observed landslide affected area. Hence, we consider a landslide as having "good" prediction when at least part of it is situated in a high or very high susceptibility zone. Otherwise, based on the above criteria, the model predicts 28 landslides in the study area, as shown in Table 3.

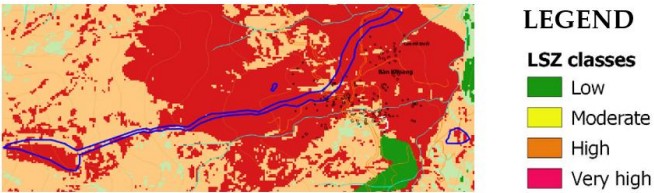

**Figure 15.** Example of some landslides overlaying the final LSZ map.

**Table 3.** LSZ validation result with observed landslide.

| Accuracy of Prediction | Observed Landslide | |
|---|---|---|
| | **Number** | **Percentage (%)** |
| Good | 22 | 78.57 |
| Wrong | 6 | 21.43 |

Table 3 indicates that 22 of the 28 observed landslides are well predicted (78.57%), and only 6 of the total landslides are wrongly predicted (21.43%). Figure 16 shows the LSZ map with the observed landslides indicating the different levels of prediction.

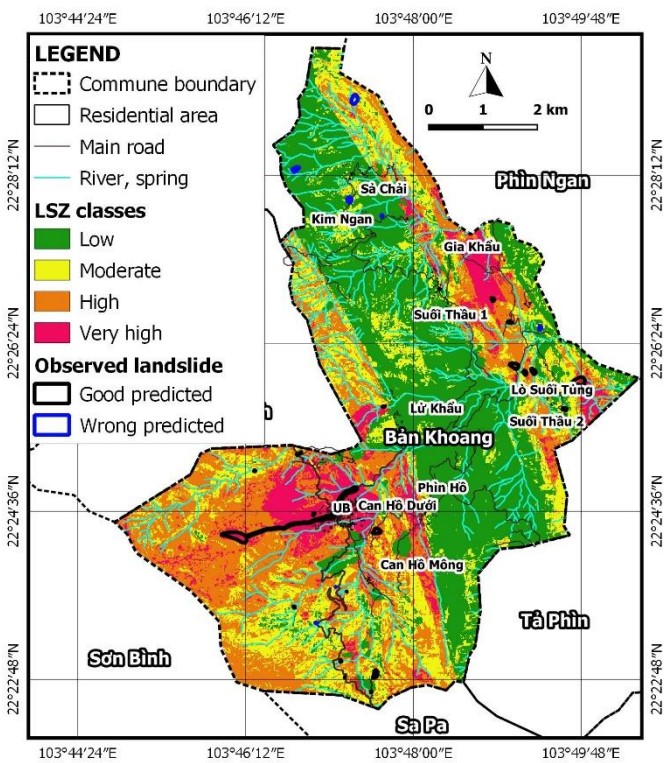

**Figure 16.** Validation LSZ map with observed landslides of Ban Khoang commune.

The Area Under the Curve (AUC) is used to qualitatively analyze the prediction accuracy of the landslide susceptibility map (Figure 17). The analysis results of the success rate curve indicated that the statistical index model has an approximately high AUC value of 0.803.

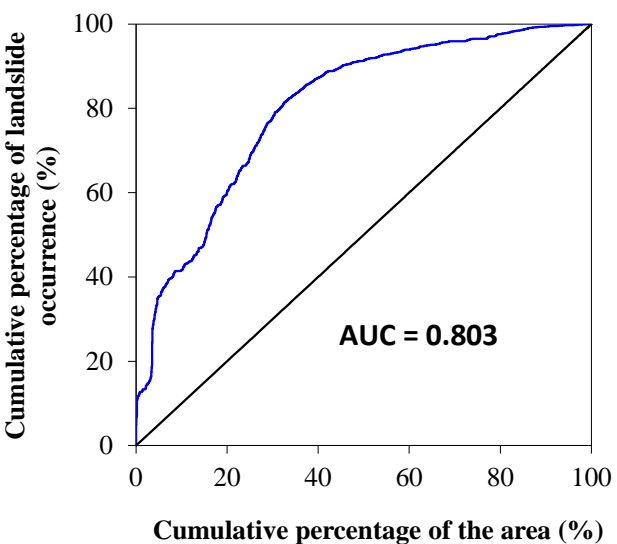

**Figure 17.** AUC representing quality model a success rate curve.

In terms of model performance, the accuracy of the statistical index method for landslide susceptibility mapping is approximately 80.3%, which is much closer to other studies (e.g., 74%

in the work of Conoscenti et al. (2016) [57], 71% in the work of Camilo et al. (2017) [58], 75% in the work of Youssef et al. (2015) [59], and 73.3% in the work of Shu et al. (2021) [60]). It must be admitted that this accuracy is not superior, which mainly includes the following reasons: one is that the data quality of the inventory is not very high, and the other is associated with the limitation of statistically based methods and assumptions of the landslide classification method.

Because we do not have such another area, a validation of the landslide susceptibility was performed as follows:

- A total of 75% of the observed landslides in the study area is selected at random (see Figure 18). These areas form the training data set. The actual selection was made arbitrarily without considering causative factors. It was only taken into account to spread the training data set as evenly as possible over the study area.
- On the basis of the training data set, a new LSZ map based on the statistical index method for the whole study area was created (see Figure 19).
- The remaining 25% of the observed landslides in the study area is used to evaluate the correctness of the new LSZ map.

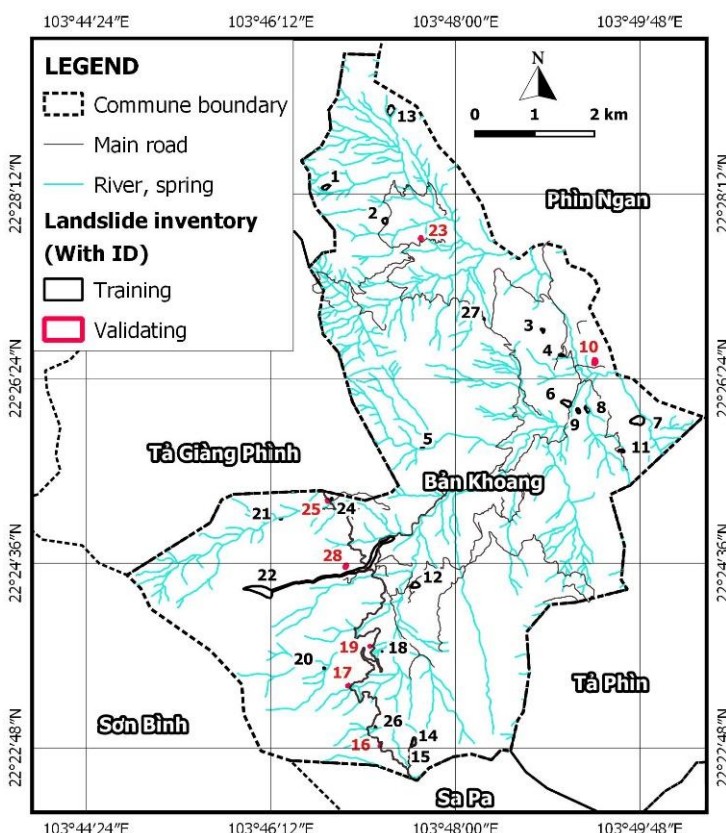

**Figure 18.** Randomly selected observed landslides in the study area for model validation.

It can be seen that the 80.95% of landslide number has "good" prediction for the new LSZ map based on 75% landslide training data set. Meanwhile, it is a higher value of "good" prediction (83.33%) for the landslide validating data set.

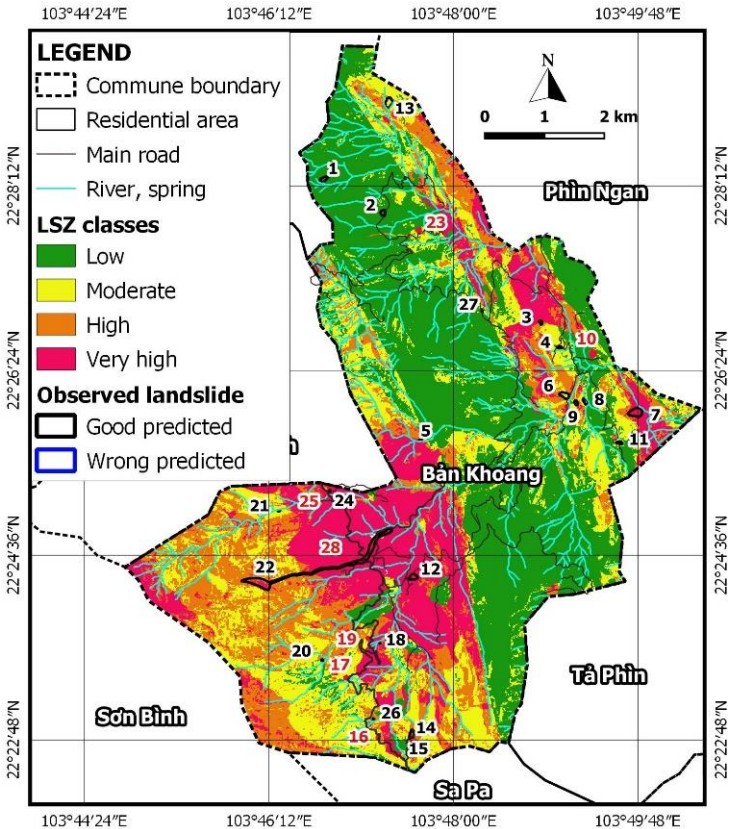

**Figure 19.** LSZ map based on 75% training landslides of Ban Khoang commune.

The area of landslide belonging to high and very high LSZ classes for the landslide training data set is 80%. Meanwhile the similar value for the landslide validating data set is obtained to be a little bit higher, with 81.2%. The model predictions for training and validating landslides in the study area are shown in Table 4. Generally, the results show that the target data can be predicted well with the modeling approach.

**Table 4.** LSZ validation result with training and validating landslide.

| Accuracy of Prediction | Landslide Training Data Set | | | | Landslide Validating Data Set | | | |
|---|---|---|---|---|---|---|---|---|
| | Number | Percentage (%) | Area (km²) | Percentage % | Number | Percentage % | Area (km²) | Percentage % |
| Wrong | 4 | 19.05 | 0.0512 | 20 | 1 | 16.67 | 0.0011 | 18.8 |
| Good | 17 | 80.95 | 0.2050 | 80 | 6 | 83.33 | 0.0047 | 81.2 |

## 7. Conclusions

Because most of the observed landslides are well predicted, the high and very high landslide susceptibility classes in the final LSZ map can be considered highly believable.

For all landslides that are wrongly predicted, of course, due to the assumptions of the landslide classification method, 6.7% and 13.3% of the total observed landslide areas fall in the low and medium landslide susceptibility class, respectively. Hence, it is easy to understand that some observed landslides are not correctly predicted.

The causes of these landslides remain unanswered in this study. This probably has to do with some unique local conditions that promote landslides that were not considered in the present analyses or errors or misinterpretations of the data and factor maps.

Finally, the good prediction can be evaluated based on observed landslides belonging to very high and high LSZ classes, it can be seen that 80% observed landslide areas and 78.57% number of observed landslides were well predicted, and AUC obtained 0.803.

Hence, the LSZ map was created, and the real-time landslide monitoring station will be reliable to use in the practice.

**Author Contributions:** Conceptualization, T.-Y.C. and L.N.T.; data curation, Y.-M.F. and H.-Y.Y.; formal analysis, L.N.T. and T.-V.H.; funding acquisition, C.-Y.L. and C.-L.W.; investigation, T.-Y.C. and C.-Y.L.; methodology, Q.D.N. and C.-L.W.; project administration, L.N.T. and Y.-M.F.; software, T.-V.H. and Y.-C.L.; supervision, T.-Y.C. and C.-Y.L.; validation, T.-V.H. and H.-Y.Y.; visualization, Y.-M.F. and Y.-C.L.; writing—Original draft, L.N.T. and T.-V.H.; writing—Review and editing, T.-V.H. and L.N.T. All authors have read and agreed to the published version of the manuscript.

**Funding:** This research received the funding from Vietnam Ministry of Natural Resources and Environment, Taiwan Soil and Water Conservation Bureau (SWCB): TNMT.2021.02.11.

**Acknowledgments:** We acknowledge the support of Vietnam Institute of Geology and Mineral Resources (VIGMR), Taiwan (Soil Water Conservation Bureau (SWCB)), GIS Research Center of Feng Chia University, WeatherPlus company. We would like to express our sincere thanks and appreciations to funded MONRE projects: "Research, design and manufacture a monitoring system for real-time early warning of landslides, debris flow, debris floods, and flash floods in mountainous and midland areas of Vietnam" code TNMT.2019.03.01 and "Study, develop a pilot debris flow early warning system in real time for mountainous areas of Vietnam" code TNMT.2021.02.11. Without the projects' supported data, it would have not been possible to complete our paper.

**Conflicts of Interest:** The authors declare no conflict of interest.

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
