# Peer review of "Using Landslide Statistical Index Technique for Landslide Susceptibility Mapping: Case Study: Ban Khoang Commune, Lao Cai Province, Vietnam"

_water, doi:10.3390/w14182814_

Round 1
Reviewer 1 Report
Table 3. LSZ validation result with observed landslide: Accuracy generated s 78.57; which is below the standard of 85% in Geospatial science.
Figure 14. Example of some landslides overlaying the final LSZ map; Author should map all the past landslides on the final LSZ. Look like the author trying to hide something.
Author Response
Dear Reviewer,
Thank you very much for your nice comments. We would like to response every points as bellow:
Point 1: Table 3. LSZ validation result with observed landslide: Accuracy generated s 78.57; which is below the standard of 85% in Geospatial science.
Response to Point 1:
The accuracy of the final LSZ map is evaluated based on the observed landslides.
Firstly, result shows that 80% observed landslide areas belonging to very high and high LSZ classes. This is totally suitable with the assumptions of the landslide classification method, 6.7% and 13.3% of the total observed landslide areas fall in the low and medium landslide susceptibility class respectively.
Secondly, the final LSZ map is checked by overlaying it with the observed landslide map. Because in the inventory of the observed landslide, no distinction was made between the initiation part of the landslide and the areas of debris or flows, there can be no complete correspondence between the LSZ classes, and the complete observed landslide affected area. Hence, we consider a landslide as “good” predicted when at least part of it is situated in a high or very high susceptibility zone. The analysis indicated that 22 of the 28 observed landslides are well predicted (78.57%), and only 6 of the total landslides are wrongly predicted (21.43%).
The AUC is used to qualitatively analyze the prediction accuracy of the landslide susceptibility map. The analysis results of the success rate curve indicated that the statis-tical index model has a approximate high AUC value of 0.803.
Because most of the observed landslides are well predicted, the high and very high landslide susceptibility classes in the final LSZ map can be considered highly believable.
For all landslides that are wrongly predicted, of course, due to the assumptions of the landslide classification method, 6.7% and 13.3% of the total observed landslide areas fall in the low and medium landslide susceptibility class respectively. Hence, it is easy to understand that some observed landslides are not correctly predicted
In terms of model performance, the accuracy of the proposed model is only approximately 80.3%, which is much closer to other studies (e.g., 74% in the work of Conoscenti et al. (2016) [[1]], 71% in the work of Camilo et al. (2017) [[2]], 75% in the work of Youssef et al. (2015) [[3]], 73.3% in the work of Shu et al. (2021)[[4]]). It must be admitted that this accuracy is not superior, which mainly includes reasons: one is that the data quality of the inventory is not very high, and the other is associated with the limitation of statistically based methods and assumptions of the landslide classification method.
Point 2: Figure 14. Example of some landslides overlaying the final LSZ map; Author should map all the past landslides on the final LSZ. Look like the author trying to hide something.
Response to Point 2:
No, nothing hides here. It was already shown in Figure 15 (old version or Figure 16 in revised version), all landslides overlaying on LSZ map. However, the picture may be so small then you can not see it.
Submission Date
23 July 2022
Date of this review
03 Aug 2022 03:38:42
[1] Conoscenti, C.; Rotigliano, E.; Cama, M.; Caraballo-Arias, N.A.; Lombardo, L.; Agnesi, V. Exploring the effect of absence selection on landslide susceptibility models: A case study in Sicily, Italy. Geomorphology 2016, 261, 222–235.
[2] Camilo, D.C.; Lombardo, L.; Mai, P.M.; Dou, J.; Huser, R. Handling high predictor dimensionality in slope-unit-based landslide susceptibility models through LASSO-penalized Generalized Linear Model. Environ. Model. Softw. 2017, 97, 145–156.
[3] Youssef, A.M.; Al-Kathery, M.; Pradhan, B. Landslide susceptibility mapping at Al-Hasher area, Jizan (Saudi Arabia) using GIS-based frequency ratio and index of entropy models. Geosci. J. 2015, 19, 113–134.
[4] Shu, H.; Guo, Z.; Qi, S.; Song, D.; Pourghasemi, H.R.; Ma, J. Integrating Landslide Typology with Weighted Frequency Ratio Model for Landslide Susceptibility Mapping: A Case Study from Lanzhou City of Northwestern China. Remote Sens. 2021, 13, 3623. https://doi.org/10.3390/rs13183623

Reviewer 2 Report
Review report
Thank you for submitting your manuscript on Using landslide statistical index technique for landslide sus-2 ceptibility mapping. Case study: Lao Cai Province, Vietnam. Therefore, I give some suggestion and question which I hope useful to the author, my decision is Major revision.
1. Introduction
It was difficult to see the justification for the need of this research. The literature review is poor. The paper needs to clearly state what are the problems with the existing works (these types of approaches) and what problem(s) this particularly paper was going to address. Without this clearly problem statement readers would have difficulty to see the merit of this paper. The author only lists some references, I did not find the problem with the exist method. The problem of the existing method is not clear. The author should show us deep analysis about the gap between existing method.
2. Methodology and data
1) Which sampling strategy had been apply to sample landslide data, why?
2) How did the author validate the method? Which method have been applied to generate the train and validate data.
3) The flowchart of the method should be added
3. Results and discussions
1) Statistical significance test of the models, and the effect of key parameter of model.
2) There were very few discussions of previous studies, the author should pay more attention or deeper analysis about the effect caused by the parameter of the model.
Other comment:
Figure: The resolution of figure should be improved.
Reference: There are a lot of latest article should be updated.
Language: The paper should be check by a native speaker.
Author Response
Dear Reviewer,
Thank you very much for your positive comments. We would like to response as bellow, please consider for us:
Review report
Thank you for submitting your manuscript on Using landslide statistical index technique for landslide sus-2 ceptibility mapping. Case study: Lao Cai Province, Vietnam. Therefore, I give some suggestion and question which I hope useful to the author, my decision is Major revision.
- Introduction
It was difficult to see the justification for the need of this research. The literature review is poor. The paper needs to clearly state what are the problems with the existing works (these types of approaches) and what problem(s) this particularly paper was going to address. Without this clearly problem statement readers would have difficulty to see the merit of this paper. The author only lists some references, I did not find the problem with the exist method. The problem of the existing method is not clear. The author should show us deep analysis about the gap between existing method.
Response:
Well noted. Please see the revised paper and give us the comments again if they are not yet sastified you.
- Methodology and data
1) Which sampling strategy had been apply to sample landslide data, why?
Response:
Basically, statistical landslide susceptibility approaches are based on related spatial information on past landslide activities (i.e. landslide presence/absence) to static geoenvironmental factors (e.g. topography, geoology, geomorphology, landuse, fault density, soil, drainage density) using statistical techniques.
Hence, the influence of sample sizes, the effect of sampling strategies or the impact of data set qualities are very important for the accuracy. Infact, a reliable landslide inventory is a vital component to achieve high-quality statistical landslide susceptibility models, also because most analysis steps are dependent on a correct representation of past landslide occurrences. Beside, the positional accuracy of an inventory is reliant on, e.g. the type and quality of the available mapping basis, time availability and the specific characteristics of landslides and the study site. In order to limit the incorrection ò landlside inventory, authors to test and check on the filedwork for observed landslide as much as we can.
2) How did the author validate the method? Which method have been applied to generate the train and validate data.
Response:
Based on your comment, we have supplemented the as the following:
- 75% of the observed landslides in the study area are selected at random. These will form the training data set. The actual selection was done arbitrarily without considering causative factors. It was only taken into account to spread the training data set as evenly as possible over the study area;
- On the basis of the training data set, a new LSZ map based on the statistical index method for the whole study area is created;
- The remaining 25% of the observed landslides in the study area are used to evaluate the correctness of the new LSZ map.
3) The flowchart of the method should be added
Response:
It’s OK. Please see the flowchart in the revised version
- Results and discussions
1) Statistical significance test of the models, and the effect of key parameter of model.
Response:
Well noted. We will add it to the revised version
2) There were very few discussions of previous studies, the author should pay more attention or deeper analysis about the effect caused by the parameter of the model.
Response:
Well noted. We will add it to the revised version.
Other comment:
Figure: The resolution of figure should be improved.
Response:
It’s OK. It will be improved in revised version.
Reference: There are a lot of latest article should be updated.
Response:
It’s OK. We updated in revised version
Language: The paper should be check by a native speaker.
Submission Date
23 July 2022
Date of this review
08 Aug 2022 04:26:11

Reviewer 3 Report
Several figures are not clear. The following revisions are necessary;
1) There is no TABLE-2. I think.
2) Line 250; Conclusions are too long. It should be divided into the two parts; One is about the accuracy of the expected landslide susceptibility mapping and another the conclusions of whole text.
3) The characters of Figures 4,5 and 15 are too small to read. Please use larger font or enlarge the figured.
4) The index for landslide susceptibility is subject to “van Westen(1997b)” If so, the short table of the relationship between the geophysical condition and the landslide susceptibility should be explained.
Author Response
Dear Reviewer,
Thank you very much for your positive comments. We would like to response as bellow, please consider for us:
Several figures are not clear. The following revisions are necessary;
- There is no TABLE-2. I think.
Response
Corrected. We will revise text in the paper
- Line 250; Conclusions are too long. It should be divided into the two parts; One is about the accuracy of the expected landslide susceptibility mapping and another the conclusions of whole text.
Response
Well noted. Please see the revision in the submited version
- The characters of Figures 4,5 and 15 are too small to read. Please use larger font or enlarge the figured.
Response
Yes, thank you. We will revise it as the comment
- The index for landslide susceptibility is subject to “van Westen(1997b)” If so, the short table of the relationship between the geophysical condition and the landslide susceptibility should be explained.
Response
Agree. Please see the explainations in the revised version
Submission Date
23 July 2022
Date of this review
15 Aug 2022 07:55:01

Reviewer 4 Report
My comments are listed below:
1. There are too many sub-figures in Figure 4. The authors should only select the necessary.
2. Table 2 is missed.
3. The text in Figures are unclear and difficult to read. The resolution of figures should also be improved and the text shoule be amplified.
4. The purpose of this study is unclear. The authors should state the purpose in introduction after literature review.
5. There are too many details in the conclusion. The conclusion should conclude the findings and key points of the study.
Author Response
Dear Reviewer,
Thank you very much for your positive comments. We would like to response as bellow, please consider for us:
My comments are listed below:
- There are too many sub-figures in Figure 4. The authors should only select the necessary.
Response:
Well noted. Maybe you are right. However, we think that if the readers can see all the landslide causative factor maps of the study area, they can easily understand and have a better visual background of the study area. Hence, we proposed to keep them.
- Table 2 is missed.
Response:
Well noted. Thank you for your comment. We will revise the paper text for correcting it.
- The text in Figures are unclear and difficult to read. The resolution of figures should also be improved and the text shoule be amplified.
Response:
Well noted. The resolution and text in pictures will be improved in the revised paper.
- The purpose of this study is unclear. The authors should state the purpose in introduction after literature review.
Response:
Well noted. Please see the revised paper.
- There are too many details in the conclusion. The conclusion should conclude the findings and key points of the study.
Response:
Well noted. Please see the revised paper.
Submission Date
23 July 2022
Date of this review
17 Aug 2022 08:47:38

Round 2
Reviewer 1 Report
Accept in present form
Reviewer 2 Report
The revised paper can be accepted.
Reviewer 4 Report
The manuscript has been improved. Most of my specific comments have been addressed.